# Impact of Surface Roughness on Ion-Surface Interactions Studied with Energetic Carbon Ions ¹³C⁺ on Tungsten Surfaces

**Maren Hellwig [1],\***, **Martin Köppen [1]**, **Albert Hiller [2]**, **Hans Rudolf Koslowski [2]**, **Andrey Litnovsky [2]**, **Klaus Schmid [3]**, **Christian Schwab [4]** and **Roger A. De Souza [4]**

[1]    Independent researchers, Heinersdorfer Str. 52, 13086 Berlin, Germany; martin_koeppen@gmx.de
[2]    Forschungszentrum Jülich GmbH, Institut für Energie—und Klimaforschung—Plasmaphysik, 52428 Jülich, Germany; a.hiller@fz-juelich.de (A.H.); h.r.koslowski@fz-juelich.de (H.R.K.); a.litnovsky@fz-juelich.de (A.L.)
[3]    Max-Planck-Institut für Plasmaphysik, Boltzmannstraße 2, 85748 Garching, Germany; klaus.schmid@ipp.mpg.de
[4]    Institut für Physikalische Chemie, RWTH Aachen University, Landoltweg 2, 52074 Aachen, Germany; schwab@pc.rwth-aachen.de (C.S.); desouza@pc.rwth-aachen.de (R.A.D.S.)
**\***    Correspondence: maren-hellwig@t-online.de

**Abstract:** The effect of surface roughness on angular distributions of reflected and physically sputtered particles is investigated by ultra-high vacuum (UHV) ion-surface interaction experiments. For this purpose, a smooth ($R_a = 5.9$ nm) and a rough ($R_a = 20.5$ nm) tungsten (W) surface were bombarded with carbon ions ¹³C⁺ under incidence angles of 30° and 80°. Reflected and sputtered particles were collected on foils to measure the resulting angular distribution as a function of surface morphology. For the qualitative and quantitative analysis, secondary ion mass spectrometry (SIMS) and nuclear reaction analysis (NRA) were performed. Simulations of ion-surface interactions were carried out with the SDTrimSP (Static Dynamic Transport of Ions in Matter Sputtering) code. For rough surfaces, a special routine was derived and implemented. Experimental as well as calculated results prove a significant impact of surface roughness on the angular distribution of reflected and sputtered particles. It is demonstrated that the effective sticking of C on W is a function of the angle of incidence and surface morphology. It is found that the predominant ion-surface interaction process changes with fluence.

**Keywords:** roughness; ion-surface interaction; angular distribution; reflection; physical sputtering; deposition; sticking; plasma-wall interaction; secondary ion mass spectrometry; nuclear reaction analysis; SDTrimSP

## 1. Introduction

Ion-surface interactions such as reflection, physical sputtering, and chemical erosion are of key importance in future nuclear-fusion devices. Nuclear fusion aims for the production of electrical power by using the fusion reaction between the two hydrogen isotopes deuterium and tritium. When the hydrogen isotopes D and T fuse, a He ion and a fast neutron are generated. The He ions as well as other impurities in the plasma are magnetically directed to the armored targets in the divertor region and undergo ion-surface interactions. The armored targets are called plasma-facing components, which will be made of tungsten in future fusion devices due to the beneficial physical and chemical properties of W. The surface of these armored targets will not be polished so that the impact of surface roughness, which will be most likely in a range of $R_a$ ~20 nm due to manufacturing processes, needs to be quantified. Since these components need to withstand extreme heat loads, they are segmented

into small tiles, also termed "castellated". Due to their castellated design the crack propagation is limited, and the formation of eddy currents is reduced. However, the segmentation of tiles leads to an enlargement of the total surface area [1]. Since the total amount of retained radioactive T needs to be limited, hydrogen retention in gaps between adjacent tiles is of concern. In present tokamaks, hydrogen is co-deposited with carbon which is eroded from other components and transported along magnetic field lines. Mixed amorphous hydrocarbon layers grow dependent on exposure conditions and period. Typical particle-surface interaction processes which are of key importance for the layer growth are particle reflection, particle sticking, physical sputtering, and chemical erosion. Experiments in the tokamaks TEXTOR and DIII-D [2] have demonstrated that mixed layers of hydrogen and carbon are located at the plasma-closest gap edges within the first mm. Since the used tungsten components have technically finished, but non-polished surfaces [2], the surface roughness is one potential reason for the observed behavior.

The interaction between a projectile and a target atom depends on the species itself, the projectile energy, and angle of incidence. For reflected and sputtered particles, the angular distribution of particles exiting the solid after the interaction is of key importance for successive interaction processes which are responsible for the particle transport into the narrow gaps. The kinematics of the interaction between a projectile and target atoms is described by successive binary collisions within the so-called binary collision approximation (BCA) [3]. The interaction in a collision is determined by the interaction potential which is typically chosen as a screened Coulomb potential [4]. Codes such as the Monte Carlo code SDTrimSP (Static Dynamic Transport of Ions in Matter Sputtering) [5] and MARLOWE [6,7] use the BCA to calculate particle and energy reflection coefficients as well as sputtering yields. Since the chemical composition of the surface and the bulk changes during particle bombardment, the fraction of reflected and sputtered particles is derived fluence dependent in SDTrimSP.

The influence of the surface morphology on the reflection of D on smooth and rough graphite was previously investigated experimentally and modeled with TRIM.SP [8]. It was found that the surface roughness has a minor influence on the reflection coefficient as a function of the angle of incidence. The influence of the surface roughness on the sputtering yields showed a larger re-deposition fraction for unpolished samples [9,10], experimentally as well as in modeling. Furthermore, it was found that the surface roughness modifies the angular dependence of the sputtering yield. Reflection coefficients and sputtering yields were determined for fusion relevant species combinations, but the angular distribution of reflected and sputtered particles of rough surfaces was not investigated so far.

To study the impact of the surface morphology on reflection and physical sputtering, dedicated ion-surface experiments are performed in a specialized ultra-high vacuum (UHV) apparatus. Furthermore, modeling of angular distributions of reflected and sputtered particles is performed with the Monte Carlo code SDTrimSP. Since SDTrimSP does not take surface morphology into account a dedicated set of pre- and post-processing routines for the given experimental surface morphology were developed [11]. The numerical and experimental methods and results are shown, compared, and the discrepancies are discussed in the following.

## 2. Materials and Methods

### 2.1. SDTrimSP Simulations

Simulations are performed with the SDTrimSP code for smooth surfaces in three dimensions (3D). To enhance the SDTrimSP results to consider rough surfaces as well, a pre- and post-processing of input and output data is performed as follows: The individual surface topology of rough surfaces induces shadowed regions where projectiles with a fixed angle of incidence cannot directly impinge. Consecutively after the interaction with the solid, the topology limits the number of possible trajectories of particles leaving the solid. Exiting particles can directly interact successively with a hilly surface morphology as depicted in Figure 1. For the simulation procedure, measured surface roughness

profiles were taken to apply realistic hilly structures. The multi-step indirect implementation of the surface roughness effect on particle-surface interaction processes is performed via following steps:

1. Pre-processing of SDTrimSP input parameters by computing a normalized angular distribution of incidence angles $\sigma_{in}(\rho_{in}(\alpha_{in}))$ with respect to the smooth surface. ($\alpha_{in}$ = fixed incidence angle to smooth surface, $\rho_{in}$ = incidence angles to rough plane, as depicted in Figure 2);
2. SDTrimSP simulations of the particle bombardment;
3. Post-processing of calculated angular distributions of particles exiting the solid by rotation of the reference system;
4. Test for shadowing: The angular distribution derived in step 3 are convoluted with a reflection probability matrix. A second interaction is treated as deposition on the surface in this approximation.

The transformation of the input data determines a normalized angular distribution of angles of incidence with respect to a global (constant) incidence angle to a smooth sample surface similar to that performed in [9,10]. As depicted in Figure 2, two reference systems are needed for the calculations. SDTrimSP uses the reference system labeled as smooth (grey lines). The second reference system relates to the rough surface (black lines) on the microscopic scale. The rough plane displayed has been determined by an atomic force microscope (AFM) [12] scan of a W sample. The incoming projectile is highlighted with a bold arrow, and two possible trajectories of reflected particles are shown by dashed-dotted lines. AFM scans of each sample surface are performed and respective angular distributions of incoming particles are computed. The surface roughness measurements were performed in three dimensions (3D). Several two-dimensional (2D) line scans were used for the calculation of the normalized angular distribution of incidence angles. It was seen that the measured surface was similar in such a way that the normalized angular distributions of incidence angles of different line scans were very similar. Thus, the surface roughness can only be approximated with SDTrimSP by considering an angular distribution of incoming particles in contrast to one incidence angle for smooth surfaces. The present procedure takes a 3D problem, approximates it in 2D and calculates a 3D output.

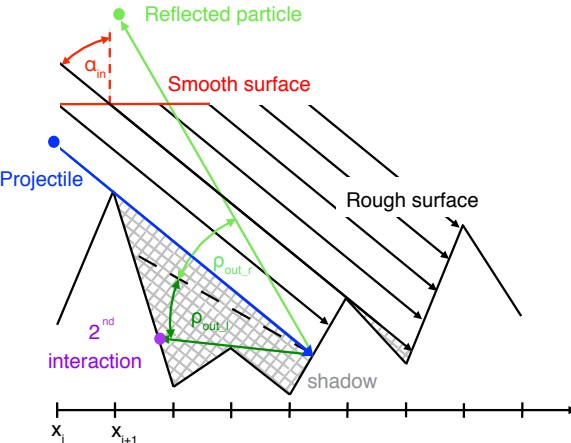

**Figure 1.** Illustration of interaction angles on rough surfaces showing a range of reflection angles which result in a second interaction.

To simulate the different experimental conditions, SDTrimSP simulations are performed with particle fluences in the range of $1.7 \times 10^{20}$ m$^{-2}$ and $10 \times 10^{20}$ m$^{-2}$, $1 \times 10^7$ test particles and a projectile energy of 950 eV. The Krypton-Carbon interaction potential [4] was used. This interaction potential defines the scattering angle of the projectile and target atom in the surface. This interaction potential approximates best the screening length in the given projectile-target constellation. Furthermore, an inelastic energy loss model which combines the Oen-Robinson and Lindhard-Scharff models [8]

was used. For smooth surfaces, the incidence angle $\alpha_{in}$ is constant, whereas for rough surfaces the normalized angular distribution of incidence angles $\sigma_{in}(\rho_{in}(\alpha_{in}))$ is used. The computed azimuthal and polar angles are taken as SDTrimSP output and transformed into spherical coordinates. The spherical coordinates are binned into intervals which corresponds to an $(2 \times 2)°$ area. This corresponds to the measurement area of the SIMS measurements as described in Section 2.3.2. Hence, computed particles are sorted according to their coordinates.

After the SDTrimSP simulations for rough surfaces, the results of the angular distributions of particles leaving the solid need to be transformed due to the angle $\gamma_{AFM}$ between the smooth surface normal and the rough surface normal by rotating the reference systems (like depicted in Figure 2).

In an additional step, the surface topology on the macroscopic scale is accounted for by calculating all possible particle trajectories in 3D for each measured slope from the AFM scan. Trajectories which do not intersect with another surface of a hilly structure, which was taken from a 2D-scan, further away from the initial interaction point are counted as being reflected with a probability of 1. Those trajectories which intersect are considered as deposited particles and are labeled with a reflection probability of 0. Thus, a reflection probability matrix of the surface is created which is used for the convolution of the modeled pre- and post-processed angular distribution of particles leaving the solid. Furthermore, particles with an azimuthal angle of $\pm 1°$ were selected of the 3D-simulation results to be comparable to the experimentally analyzable area of the secondary ion mass spectrometry (SIMS) measurements. In the following, the conduced ion-surface experiments will be described.

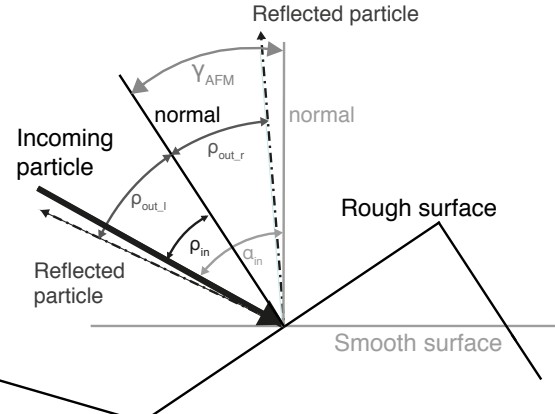

**Figure 2.** Definition of angles and reference systems for the pre- and post-processing: $\alpha_{in}$ represents the fixed incidence angle to the smooth surface, $\rho_{in}$ the angles of incidence to the rough plane and $\Gamma_{AFM}$ the angle which is measured with an atomic force microscope (AFM). The angles $\rho_{out\_l}$ and $\rho_{out\_r}$ represent possible exit angles of one reflected particle. In this picture $|\rho_{out\_l}| = |\rho_{out\_r}|$. For a cosine angular distribution with its center around the surface normal, both angles have the same likelihood.

## 2.2. Experiments

Investigations of the influence of the surface morphology on the angular distribution of reflected and sputtered particles are performed with $C^+$ ions impinging W surfaces at 950 eV. Projectiles at the given energy will uniformly bombard the sample surface in contrast to projectiles with energies close to the surface binding energy. It the latter case, the surface morphology with its slopes and sinks could affect the interaction process more strongly, which needs to be investigated in a different study.

C is chosen, since it is one of the main plasma impurities in present-day tokamaks and stellarators. To distinguish C atoms resulting from the individual ion-surface interaction processes from typical surface contamination, $^{13}C^+$ is used due to its low natural isotope abundance of 1.11%. Experiments are performed by bombarding smooth and rough W samples with 950 eV $^{13}C^+$ ions at incidence angles of 30° and 80° to the surface normal. Reflected and sputtered $^{13}C$ is captured on a cylindrically bent Ti foil for each of the four experiments individually. Exposed Ti foils are analyzed by means of SIMS.

In addition, the areal density of deposited $^{13}$C on the W surface is quantified by nuclear reaction analysis (NRA).

### 2.2.1. Experimental Setup

Experiments were performed in a self-built UHV apparatus ALI [13], schematically depicted in Figure 3. The ion source ionizes the gas by electron impact ionization. After the extraction, an ion lens (1) collimates the ion beam to a spot with a diameter <1 mm. Consecutive deflector plates enable the alignment of the beam in two dimensions. The ions are mass selected in a magnetic sector field. A dedicated system of electrostatic lenses (2) and deflection plates (3) subsequently focuses the ions onto the target. Samples are mounted on a rotatable manipulator system from VG Scienta which is inserted into the vacuum recipient from the top flange. To compute the accumulated fluence during experimental operation, the sample current is recorded.

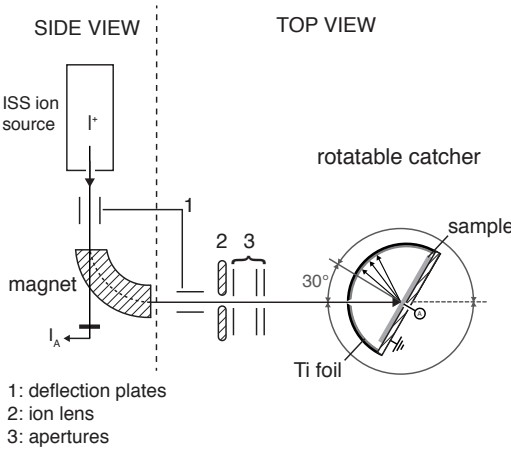

**Figure 3.** Illustration of the ultra-high vacuum ion beam experiment based on [13].

### 2.2.2. Sample Preparation

The surfaces of the W samples of 10 mm × 18 mm × 1 mm were prepared by chemical etching. This process yields the so-called *rough* samples. HNO$_3$ and HF were used as the etching medium. The *smooth* samples were additionally polished mechanically in several steps by using *FEPA P* silicon carbide paper (FEPA, Paris, France) *P800* (21.8 µm ± 1 µm), *P1200* (15.3 µm ± 1 µm), *P2500* (8.5 µm ± 1 µm), *P4000* (6 µm) followed by a diamond suspension with grains of 3 µm size, and a diamond suspension with 1 µm grains. Finally, the surface was polished with OP-S solution, a colloidal silica suspension with grains of 0.04 µm size. Microscopic images of both sample types are shown in Figure 4. The surface topology is measured with an Agilent 5400 AFM (Agilent Technologies, Santa Clara, CA, USA) on 100 µm × 100 µm areas, which was the maximum possible scanning area of the used AFM. The resolution of the measurements is 195 nm which equals 512 × 512 measured points. The scanned areas represent the sample morphology. For the simulations, AFM measurement results of lines scans are used and are depicted in Figure 5. The arithmetic ($R_a$), ten-point ($R_{10}$), and root-mean square ($R_{rms}$) roughness were calculated for both topology types [14] and are summarized in Table 1.

The Ti catcher foils of 125 µm thickness were prepared by cleaning the surface with a radio-frequency Ar plasma in the PADOS device [15]. Thus, surface impurities descending from the manufacturing process and due to air exposure are eliminated. After cleaning, the samples were stored under vacuum to reduce new deposition due to air exposure. It was shown by performing NRA measurements of cleaned and non-cleaned Ti foils that the cleaning process led to a reduction of $^{13}$C by 10 times. Titanium was chosen due to a preferable combination of an easy bending of the foil, the interacting species, and the used surface analysis methods.

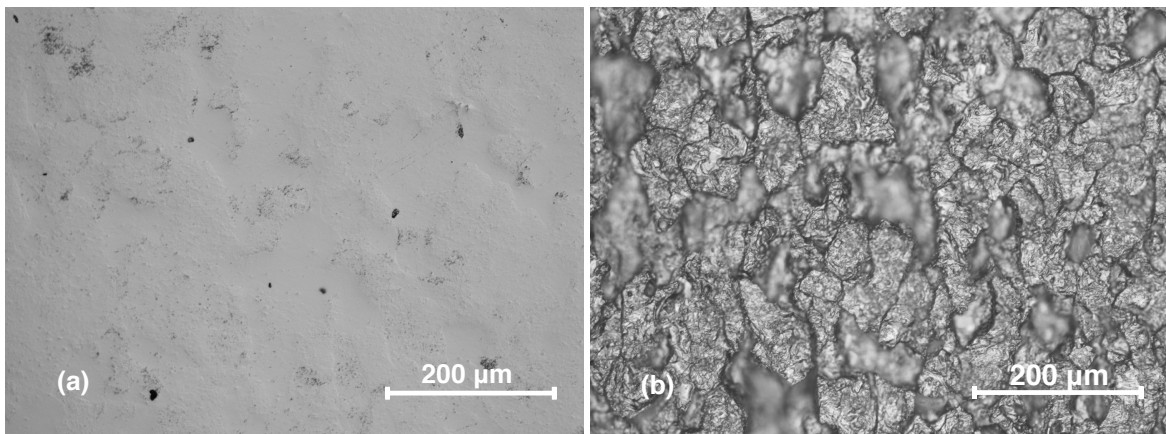

**Figure 4.** Images of the smooth (**a**) and rough (**b**) W sample taken with an optical microscope.

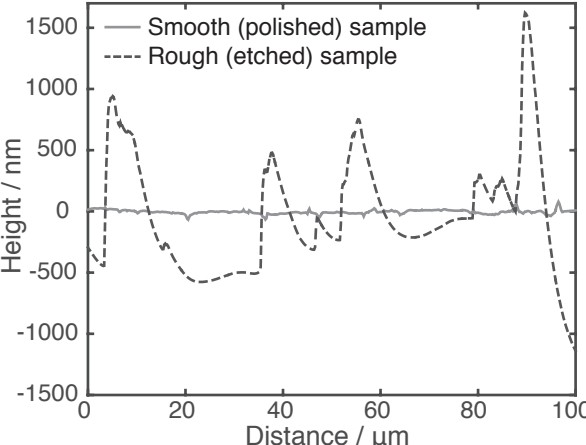

**Figure 5.** The AFM profiles of the surface topology of a smooth/polished (solid line) and a rough/etched (dashed line) W sample.

**Table 1.** Roughness parameters of the polished and etched W samples: The arithmetic ($R_a$), ten-point ($R_{10}$), and root-mean square ($R_{rms}$) roughness were calculated for both topology types with formulas from [14].

| Surface Type | $R_a$ [nm] | $R_{10}$ [nm] | $R_{rms}$ [nm] |
|---|---|---|---|
| Smooth | 5.93 | 348.52 | 43.70 |
| Rough | 20.53 | 333.13 | 569.82 |

### 2.2.3. Target and Catcher Design

The target and catcher (Figure 6) are designed to allow an interaction of the projectile beam with the W surface and to collect the reflected and sputtered $^{13}$C atoms on the Ti catcher foil in an azimuthal plane of $0°$. The catcher as well as the mounted Ti catcher foil are on ground potential and have a radius of 6 mm. Possible incidence angles of the projectile beam are defined by entry apertures in the catcher and start from normal incidence with $0°$, $30°$, $45°$, $60°$, and $80°$ with respect to the surface normal. For each experiment, a Ti foil with a hole aligned at one of the entry aperture positions is prepared. The W sample is insulated from the catcher with a Kapton foil (hatched) to allow monitoring of the ion beam current.

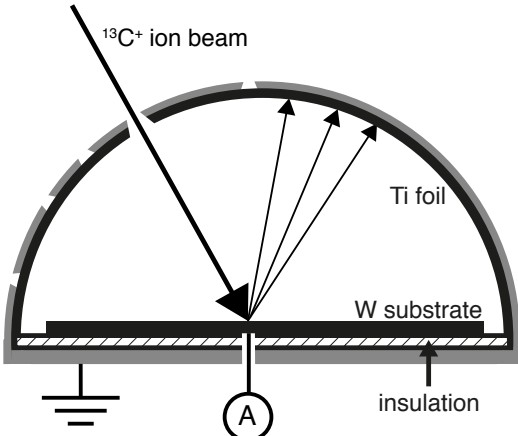

**Figure 6.** Illustration of the rotatable catcher of the ion-surface interaction experiment. Apertures at $0°$, $30°$, $45°$, $60°$, and $80°$ with respect to the surface normal are visible in the catcher setup (grey). For the experiments individual Ti foils are taken with a hole aligned at one of the entry aperture positions as shown for $30°$.

### 2.2.4. Experimental Parameters

For each experiment $^{13}C$ ions were produced by dissociative electron impact ionization of $^{13}CO$. The W surface was bombarded up to 450 h to accumulate the $^{13}C$ deposition necessary for the post-mortem analysis of the Ti foil. The steady-state beam bombardment was controlled by using a high precision thermomechanical valve from Pfeiffer vacuum with thermomechanical control of the gas flux into the source. A base pressure below $5 \times 10^{-8}$ hPa is achieved in the recipient leading to a mean free path for carbon ions of the order of $10^6$ m. The accumulated particle fluence was determined by measuring the sample current as a function of time in combination with a determination of the spot area after the exposure.

### 2.3. Analyses

The bombarded W surfaces as well as the Ti foils are analyzed with respect to the deposited C. An absolute, quantitative analysis of the deposited C on the W surface as well as the reflected and sputtered C collected on the Ti foils is crucial to understand the underlying ion-surface interaction processes. By combining a NRA of W surfaces with an analyses with SIMS of the Ti foils, it is possible to compare absolute values of the conducted experiments and thus to isolate important mechanisms in the following.

### 2.3.1. NRA

The $^{13}C$ deposition on the W samples is analyzed with NRA [16] to determine quantitatively the areal density of $^{13}C$. The measurements were performed with the reaction $^{13}C(^3He,p_0)^{15}N$. A $^3He$ beam with 3.15 MeV and a spot diameter of 1 mm is used to trigger nuclear reactions with the $^{13}C$ deposition on the W samples. RBS (Rutherford backscattering spectrometry) and NRA spectra are measured under a scattering angle of $165°$ for two samples and two spots on each sample. The areal density of $^{13}C$ is determined with the SIMNRA program [17] by fitting experimentally obtained spectra to simulated spectra. A normalized particle balance for each experiment is estimated via the ratio of the deposited $^{13}C$ and the corresponding accumulated fluence on the sample.

### 2.3.2. SIMS

The angular distribution of reflected and sputtered $^{13}$C is quantified by multiple SIMS [18] measurements on the Ti catcher foils. SIMS measurements are performed in fashion of a line scan along the median of the Ti foil at an azimuthal angle of 0° with a *Time-of-Flight SIMS IV* machine (IONTOF GmbH, Münster, Germany) at the RWTH Aachen University. Single measurement areas are scanned with a 2 keV Cs$^+$ sputter beam over an area of 200 μm × 200 μm and a 25 keV Ga$^+$ analysis beam which scanned 81.3 μm × 81.3 μm inside the sputtered area to eliminate edge effects. The analysis beam focus is adjusted for each measurement since the Ti foil remained slightly bent due to the experimental setup, although the bending of the Ti foil is minimized by the holder setup.

As mentioned above, absolute values are needed to compare the four conducted experiments. Thus, in the following the evaluation applied is described in detail since it deviates significantly from the standard evaluation. In the SIMS analysis the $^{13}$C to $^{12}$C ratio is estimated. Hydrocarbons are found and need to be considered additionally since $^{13}$CH shows a non-proportional behavior to $^{12}$CH into the depth of the sample. This is attributed to the foil exposure in ALI. Special care is taken during the measurement to optimize the separation of the $^{13}$C and $^{12}$CH as well as the $^{12}$CH$_2$ and $^{13}$CH mass peaks. To separate the isobaric interferences correctly, a non-standard analysis method is applied. The analysis procedure consists of the following steps, which will be described in the following:

1. Reconstruction of mass spectra for recorded time steps.
2. Dead time correction of mass peaks $^{12}$C, $^{12}$CH, $^{13}$C, $^{12}$CH$_2$, and $^{13}$CH.
3. Peak fitting of $^{12}$C, $^{12}$CH, $^{13}$C, $^{12}$CH$_2$, and $^{13}$CH.
4. Determination of the depth profiles of individual species.
5. Background correction of the $^{13}$C isotope measurement.

The data reconstruction and dead time correction is performed with the software Surface Lab 6.5 (ION-TOF GmbH, Münster, Germany) [19]. For the deconvolution of isobaric interferences of mass 13 and 14 the software CasaXPS 2.3.16 PR 1.6 (Casa Software Ltd., Teignmouth, UK) [20] is used. Especially for the first few analysis cycles the peak fitting results demonstrated severe differences to the standard SIMS method. This is caused by a strong peak asymmetry with a pronounced tail on the right side as depicted in Figure 7a. The peak shape originates from the measurement setup which causes an angular aberration and from calibration factors such as the high reflector voltage of U = −50 V which causes a spread in the arrival time of secondary ions in the detector [21]. Each side of the mass peak is thus fitted with a Lorentz distribution [20] with a full-width-half-maximum (FWHM) $f$ and the peak position $e$ with $x \leq e$ for the left side and $x > e$ for the right side:

$$L_{asym}\left(x : \alpha, \beta, f, e\right) = \begin{cases} \left[L\left(x : f, e\right)\right]^{\alpha} & x \leq e \\ \left[L\left(x : f, e\right)\right]^{\beta} & x > e \end{cases} \tag{1}$$

so that $L_{asym}$ describes the left-to-right peak asymmetry. For continuity both Lorentz distributions are convoluted with a Gaussian distribution. In the present case $\alpha > \beta$ so that the asymmetric tail to the right side is reproduced. The exact peak shape is determined by fitting the non-disturbed $^{12}$C peak. Shape shifting [22,23] to other peaks is applied in the fitting process to each time step for a single measurement crater. The integration of the fitted curves yields the depth profiles of the individual masses.

The background correction is required due to the natural $^{13}$C abundance. Figure 7c shows that the Ti foil is contaminated with $^{12}$C and $^{13}$C throughout the analyzed layers. The $^{13}$C deposition originating from the experiment is present in roughly the first 50 sputter seconds which corresponds to the depth. The intensity is corrected in a multi-step procedure. The minimum of the summed intensity counts of $^{13}$C and $^{13}$CH is computed ($k$) and serves as the start depth to calculate the mean ratio of background counts $\langle b \rangle$ for time steps $i$ and the maximum number of single measurements $N$ with

$$\langle b \rangle = \frac{1}{N} \sum_{i=k}^{N} \frac{I_{13C,i} + I_{13CH,i}}{I_{13C,i} + I_{13CH,i} + I_{12C,i} + I_{12CH,i} + I_{12CH_2,i}} \tag{2}$$

$$\sigma(b) = \frac{1}{N} \sum_{i=k}^{N} (b_i - \langle b \rangle)^2 \tag{3}$$

at each measurement position individually. The natural abundance level of $^{13}C$ amounts to 1.11%. The experimentally determined abundance level of $^{13}C$ has a maximum error interval of (0.7–2.2)% which is a result of a small deviation in the analysis beam focus. The background intensity counts $(I_{13C} + I_{13CH})_{Bkg}$ are computed for each sputter time step $i$ by

$$\left[ (I_{13C} + I_{13CH})_{Bkg} \right]_i = \frac{\langle b \rangle}{1 - \langle b \rangle} \cdot \left[ I_{12C,i} + I_{12CH,i} + I_{12CH_2,i} \right] \tag{4}$$

as a fraction of the intensity counts of $^{12}C$ and its hydrocarbons. The background $(I_{13C} + I_{13CH})_{Bkg}$ is subtracted from the signal $(I_{13C} + I_{13CH})$ in order to obtain the $^{13}C$ deposition $(I_{13C} + I_{13CH})_{Exp}$ resulting from reflection and self-sputtering during the ion-surface interaction experiments for each time step $i$ with

$$\left[ (I_{13C} + I_{13CH})_{Exp} \right]_i = \left[ I_{13C,i} + I_{13CH,i} \right] - \left[ (I_{13C} + I_{13CH})_{Bkg} \right]_i . \tag{5}$$

An example of the background corrected $^{13}C$ depth profile is shown in red in Figure 7c.

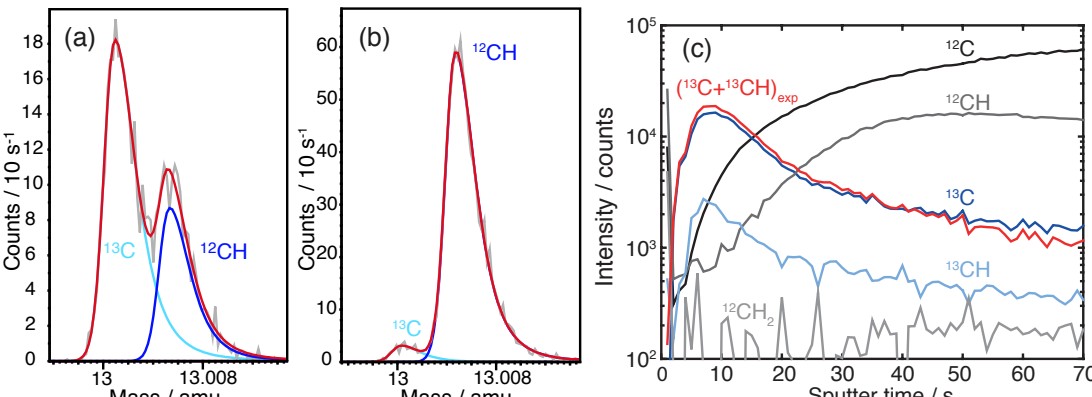

**Figure 7.** Extraction of the C intensities via peak fitting with CasaXPS [20] to reconstruct the corresponding depth profile for each SIMS crater. As an example the deconvolution of two mass spectra is shown. Mass spectrum (**a**) is taken after the 9th sputtering cycle and mass spectrum (**b**) after the 69th sputtering step. The mass spectrum illustrates the convolution of the $^{13}C$ and CH mass peaks in different sample depths. The light and dark blue curves show the $^{13}C$ and $^{12}CH$ fit, respectively. The red curve represents the envelope curve. Graph (**c**) depicts the results of the SIMS analysis for one crater. Each curve shows the contribution of the respective component to the mass spectrum as a function of sputter time.

### 3. Results

The results of the SDTrimSP simulations regarding effective reflection coefficients and sputtered fractions as a function of the fluence are shown and discussed. Furthermore, the experimentally obtained angular distributions are compared with corresponding simulations.

#### 3.1. Incidence Angle of 30°

$^{13}C^+$ bombardment of a W surface with $950\,eV$ leads to a composition change of the surface which is fluence dependent. For a smooth surface, the calculated fraction of reflected $^{13}C$ as well as the fraction of sputtered W is reduced by 0.03 as depicted in Figure 8a. A small increase of the self-sputtered fraction of $^{13}C$ up to 0.01 due to the deposition of $^{13}C$ during the exposure is observed. For the rough surface (Figure 8b) a significant different behavior is seen in comparison to the smooth surface. A greater amount of $^{13}C$ is deposited on the surface and the effective reflection decreases by 0.20. The sputtering of W decreases by 0.20 as well. The dominating process changes as a function of the fluence. In the last quarter of the experimental time which corresponds to particle fluence above $5.5 \times 10^{20} m^{-2}$ in this plot, self-sputtering of $^{13}C$ gets dominant compared to the decreasing reflected fraction of $^{13}C$ and sputtered fraction of W. At a fluence of roughly $7 \times 10^{20} m^{-2}$ self-sputtering occurs in 43% of the $^{13}C$ interactions.

Figure 9a displays the angular distributions obtained for the smooth surface. The angle of incidence corresponds to $-30°$. All particles which are reflected or sputtered with a negative velocity component $v_y$ are shown at negative exit angles. Thus, all particles with a positive velocity component are depicted with positive exit angles since they are directed in forward direction. The experimental data which is derived via the SIMS analysis routine is depicted in red squares. A mainly specular reflection at $+30°$ is observed experimentally. The corresponding SDTrimSP simulation is shown in black. In comparison to the grey curve, which represents a cosine distribution, the simulation results show a similar broad, almost cosine angular distribution. Considering only interactions with collision cascades with less than (i) 20 collision (green) or (ii) less than 10 (blue) collisions in a cascade, a shift of the distribution maximum is observed. Thus, the location of the maximum depends on number of collisions in a cascade.

Figure 9b shows the angular distributions obtained when a rough surface topology is bombarded. The experimentally obtained distribution (red squares) is a symmetric distribution due to the dominant self-sputtering process. The calculated angular distribution which considers all collisions shows a broad, almost cosine distribution in black. Selecting the maximum number of collisions (in green and blue) does not show a change of the location of the maximum with the number of collisions in a collision cascade.

Furthermore, the NRA analysis of the smooth and rough W samples reveal a similar deposited $^{13}C$ areal density $D_{NRA}$ with $D_{NRA,rough}/D_{NRA,smooth} \sim 0.92 \pm 0.08$. This result is compared to the SDTrimSP results of the integrated sputtered and reflected fractions: Although 34% less $^{13}C$ is reflected from the rough surfaces in comparison to the smooth surface over the experimental duration, a roughly equal amount of ejected $^{13}C$ is calculated for the rough surface. The integrated $^{13}C$ amount which is ejected by self-sputtering is a factor of 4.7 higher for the rough surface and thus compensates for the integrated smaller reflected fraction of $^{13}C$ of the rough surface. Therefore, an equivalent amount of $^{13}C$ is reflected ($A_{SDTrimSP}$) as well as deposited on the rough and smooth W sample for an incidence angle of 30° with $A_{SDTrimSP,rough}/A_{SDTrimSP,smooth} \sim 0.98$. The SDTrimSP results agree with the NRA analysis results for the ion-surface interaction experiment at an incidence angle of 30°.

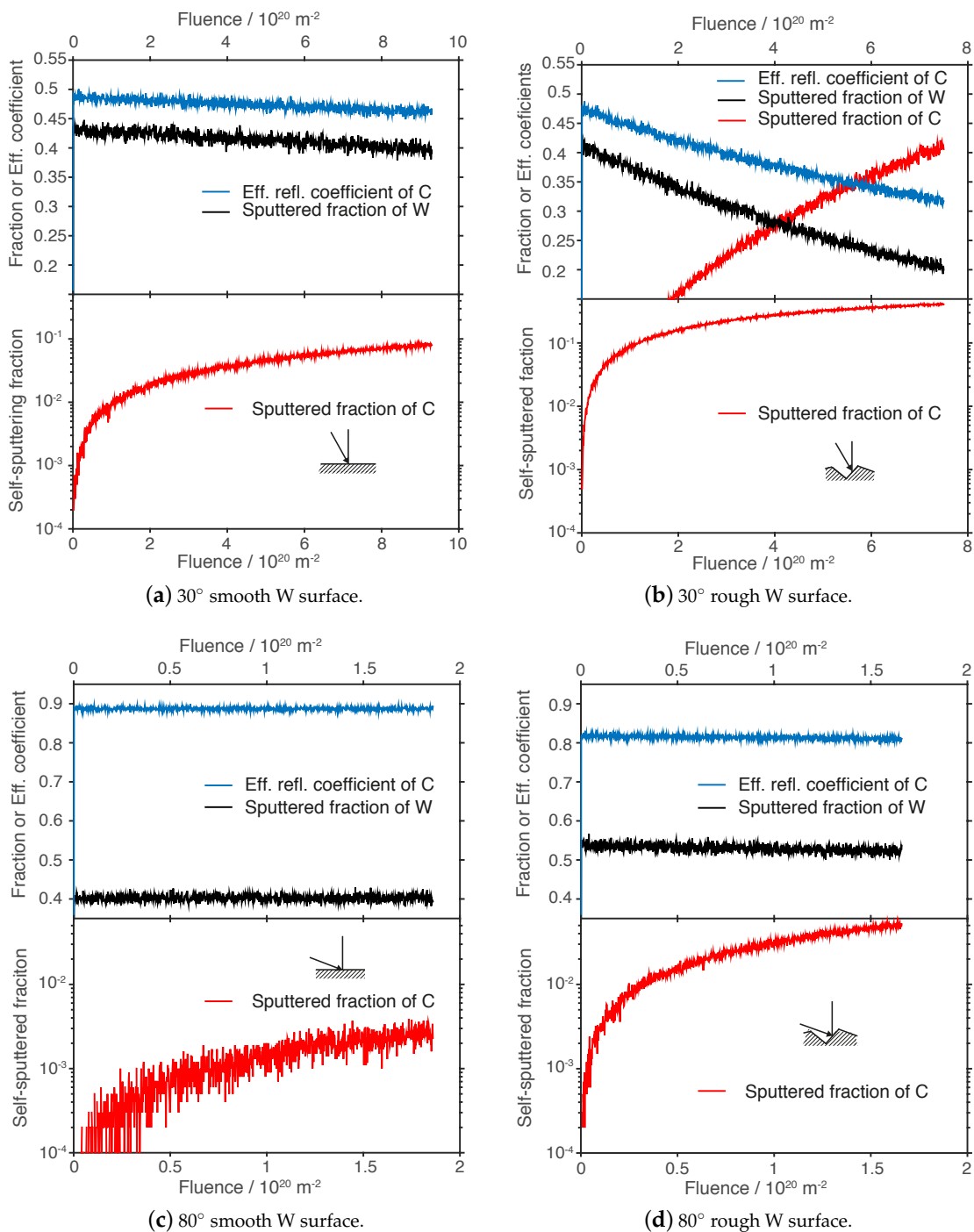

**Figure 8.** The fluence dependence of sputtered fractions and effective reflection coefficients computed by SDTrimSP is shown for different angles of incidence and surface morphology. The upper graph of each subplot illustrates the effective $^{13}$C reflection coefficient and the sputtered fraction of W. The respective lower graph shows the self-sputtered fraction of $^{13}$C which is plotted on a logarithmic scale for better visibility (In case (**b**) the self-sputtered fraction of $^{13}$C is also drawn in the upper graph). (**a**,**b**) show the bombardment of $^{13}$C under an incidence angle of 30° on a smooth and rough W surface, respectively. (**c**,**d**) show the bombardment of $^{13}$C under an incidence angle of 80° on a smooth and rough W surface, respectively.

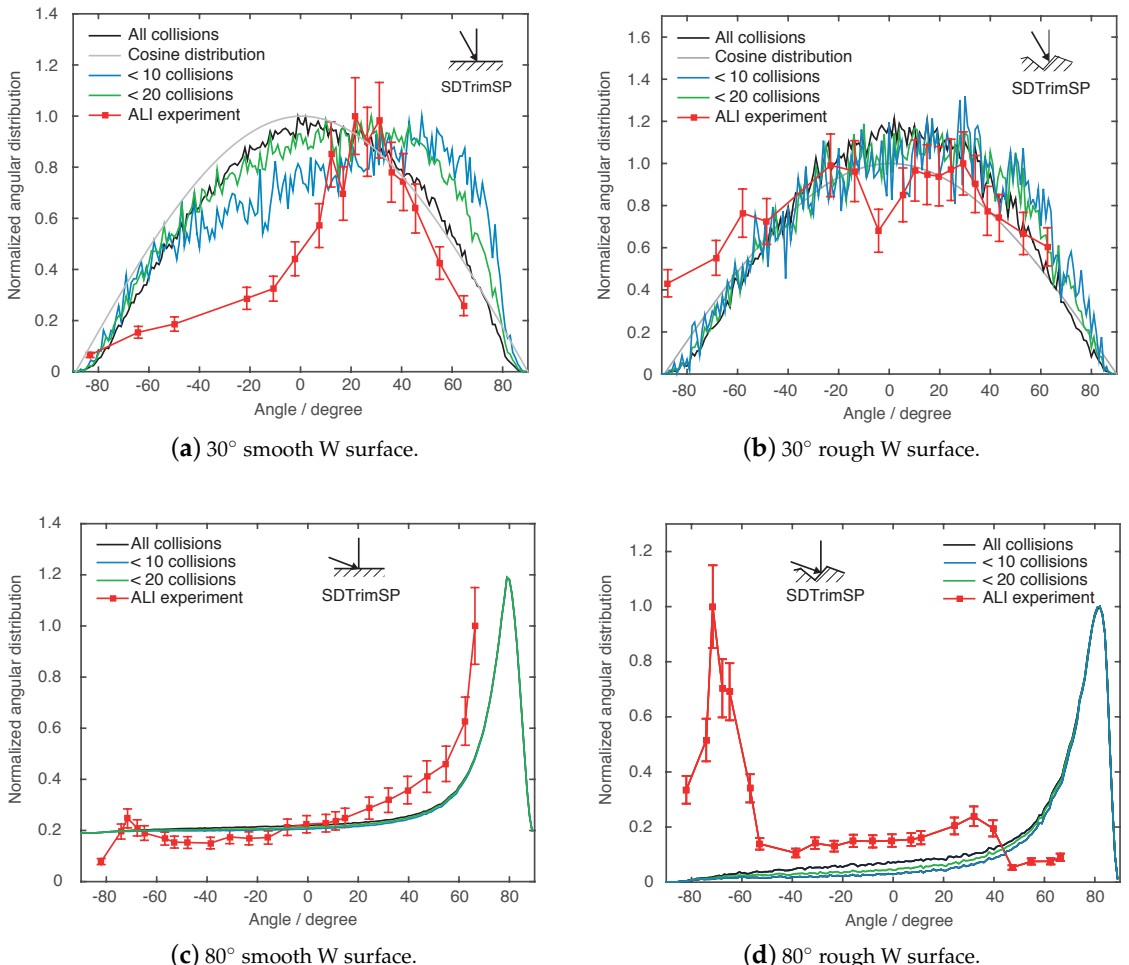

**(a)** 30° smooth W surface.

**(b)** 30° rough W surface.

**(c)** 80° smooth W surface.

**(d)** 80° rough W surface.

**Figure 9.** Comparison of experimentally and modeled angular distributions of $^{13}C$ ions impinging on W surfaces. The experimental data is depicted with red squares. The modeled data is illustrated in black, grey, green, and blue lines. Graph (**a**) shows results for the incidence angle of 30° on a smooth W sample. Graph (**b**) depicts the results for an incidence angle of 30° on a rough W sample. Graph (**c**) displays the results at an incidence angle of 80° on a smooth W sample and graph (**d**) the results of the bombardment at an incidence angle of 80° on a rough W sample.

### 3.2. Incidence Angle of 80°

Ions which bombard a smooth and rough surface under an incidence angle of 80° show a completely different behavior of reflected and sputtered fractions as well as angular distributions. The fraction of reflected $^{13}C$ from a smooth W surface (Figure 9c) is about 0.90 and remains steady. In addition, W is sputtered with a fraction of 0.40 at a constant level. Thus, the surface composition does not change as drastically as observed at an incidence angle of 30° under similar conditions, so that self-sputtering of $^{13}C$ has a sputtering yield of 0.003 for a smooth surface. For rough surfaces still a fraction of 0.82 of the $^{13}C$ is reflected and fraction of 0.53 of W is sputtered (Figure 8d). Both fractions decrease only slightly about 0.02 up to maximum fluence. The self-sputtered fraction of $^{13}C$ does increase more rapidly up to a fraction of 0.05 compared to the smooth case.

Figure 9c depicts the angular distribution of reflected and sputtered $^{13}C$. The experimentally obtained distribution (red squares) shows a mainly specular reflection in forward direction. Due to technical limitations it was not possible to access data points beyond 70°. The simulation exhibits a similar distribution which shows specular reflection with a maximum at 78° (black curve). Filtered simulation results with a selected maximum number of collisions in a collision cascade are plotted for

(i) less than 20 (green) and (ii) less than 10 (blue) collisions. It can be seen that the location of maximum is not correlated with the number of collisions in a cascade for the given case.

Figure 9d shows the angular distributions which are obtained by the bombardment of a rough surface. The experimentally obtained distribution (red squares) demonstrates mainly backward reflected $^{13}$C with a maximum at $-70°$. Calculated angular distributions however show a specular reflection in forward direction with a maximum at $+78°$. A limitation of the maximum number of collisions in a collision cascade does not lead to significant differences in the distribution. Again, no correlation of the maximum of the distribution with the number of collisions in a cascade is found.

The NRA analysis of the W samples exhibit that for particles bombarding the rough and smooth sample at incidence angle of $80°$ a roughly five times greater areal density of $^{13}$C ($D_{NRA, rough}/D_{NRA, smooth} \sim 4.81 \pm 0.08$) is found on the rough surface. The SDTrimSP results of the integrated sputtered and reflected fractions indicate a similar trend: A 18% greater amount of $^{13}$C is reflected from the smooth surface in comparison to the rough surface. In addition, the self-sputtered fraction of $^{13}$C from the smooth surface is 5% of the self-sputtered fraction $^{13}$C from the rough surface. The simulation results suggest in total a ratio of the ejected $^{13}$C of $A_{SDTrimSP, rough}/A_{SDTrimSP, smooth} \sim 0.84$ which leads to a 1.19 greater deposited $^{13}$C amount on the rough sample. Although the simulation results are not in agreement with the numbers of the NRA analysis results, a trend towards a greater deposition on the rough surface is demonstrated.

## 4. Discussion

It is observed that the number of collisions in the collision cascade is important with respect to the agreement between experimental and calculated angular distributions. Figure 10a illustrates the energy distribution of reflected and sputtered particles leaving the surface. Figure 10b shows the distribution of the number of collision of particles leaving the surface. The numerical values of these figures are compiled in Table 2. For an incidence angle of $30°$ the energy distribution is broad with a maximum at roughly 700 eV and an average of 500 eV for a smooth and rough surface. The average numbers of collisions in a cascade is calculated to amount to roughly 19 collisions with an average pathlength of roughly 45 nm before the particle exists the surface.

At an incidence angle of $80°$ the energy distribution shows that particles leave the solid with an average of 758 eV for the rough surface and 815 eV for the smooth surface. The maximum of the energy distribution is at roughly 900 eV. As a consequence, particles which interact with the solid collide on average about 7 times with target atoms with an average pathlength of 14–20 nm. The corresponding distribution of the number of collisions has its maximum around 4 collisions.

Additionally, to the SDTrimSP simulations, calculations were performed with the MARLOWE code [6,7,24] which also uses the BCA to compute particle-surface interactions. Differently to SDTrimSP, MARLOWE takes a crystalline solid into account. SDTrimSP results for the length of the collision cascades as well as the distributions are confirmed with MARLOWE simulations for the smooth surface (Table 3). The artifact, shown as a step in the energy distributions at 700 eV of the SDTrimSP simulations in Figure 10a is found with MARLOWE calculations as well. The reason for the artifact needs to be investigated further and is most likely generated by the implemented energy loss approximations in both codes. This investigation is beyond the scope of the presented study but must be pursuit further. In general, a larger angle of incidence leads to smaller number of collisions in a collision cascade (for $30°$: 19 collisions; for $80°$: 7 collisions) and higher exit energies of reflected particles.

The SDTrimSP results show a correlation between the number of collisions in a collision cascade and the corresponding shape of the angular distribution of reflected particles. Particles which are reflected after less than 6 collisions show a more specular angular distribution. Thus, the momentum components parallel to the surface is not significantly changed. Particles which are reflected after more than 6 collisions in a collision cascade are more likely to be ejected with a cosine distribution.

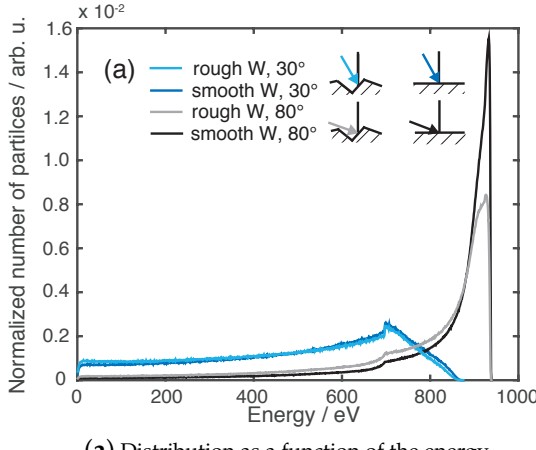
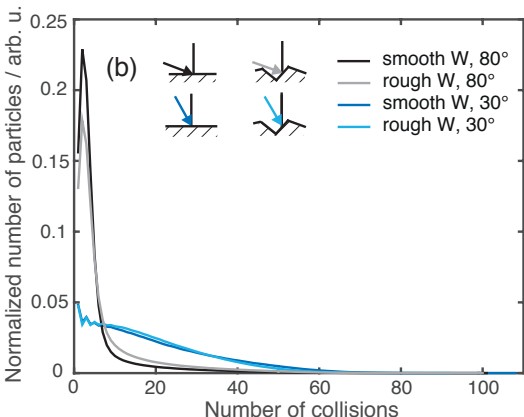

(**a**) Distribution as a function of the energy.　　　　(**b**) Distribution as a function of the number of collisions.

**Figure 10.** Normalized distributions of (**a**) the energy of the reflected and sputtered particles and (**b**) the number of collisions in the collision cascade for incidence angles of 30° and 80° calculated for smooth and rough W surfaces.

**Table 2.** Summary of the energy, number of collisions, and collision cascade length from reflected and sputtered $^{13}$C projectiles which are calculated via SDTrimSP. Each simulation is performed with $10^7$ particles, the number of particles which exit the target are shown as well.

| Surface Type | Incidence Angle/° | Average Energy/eV | Max. # of Collisions | Average # of Collisions | Average Path-Length/nm | Max. Path-Length/nm | Particles Exited Solid |
|---|---|---|---|---|---|---|---|
| Smooth | 30 | 500 | 108 | 19 | 47.6 | 266.6 | $4.7 \times 10^6$ |
|        | 80 | 815 | 101 | 6 | 13.5 | 249.5 | $8.8 \times 10^6$ |
| Rough  | 30 | 476 | 92 | 18 | 43.2 | 224.7 | $3.9 \times 10^6$ |
|        | 80 | 758 | 100 | 8 | 19.6 | 240.1 | $8.1 \times 10^6$ |

**Table 3.** Summary of the energy and number of collisions from reflected $^{13}$C projectiles which are computed with MARLOWE for a smooth W surface. For the simulation of the 30° case $5 \times 10^6$ particles and for the 80° case $2 \times 10^6$ particles are considered.

| Incidence Angle/° | Average Energy/eV | Max. Energy/eV | Average # of Collisions | Max. # of Collisions | Particles Exited Solid |
|---|---|---|---|---|---|
| 30 | 517 | 875 | 16 | 128 | $1.6 \times 10^6$ |
| 80 | 809 | 913 | 5 | 178 | $1.8 \times 10^6$ |

Furthermore, it is demonstrated that challenges arise from the comparison between experiment and simulation. One possible contribution to the found discrepancies is the approximation of a 3D surface morphology to a 2D problem for the input of SDTrimSP. Thus, 3D simulations with the 3D morphology should be performed in future studies. While particles, in the present study, impinging at an incidence angle of 30° on a rough W surface and 80° on a smooth surface, experimental and numerical angular distributions are in rather good agreement. For the case of particles impinging at an incidence angle of 30° on a smooth W surface and at an incidence angle of 80° on a rough W surface, discrepancies are observed and discussed in the following.

For particles impinging with an incidence angle of 30° on a smooth W surface, the experiment shows a higher specular reflection than observed in modeling. Particles in the simulation have a greater number of collisions and are ejected with a cosine distribution. Thus, by comparing experimental and numerical data, the location of the maximum of the angular distribution is correlated with the number of collisions in the collision cascade.

For $^{13}$C ions which impinge a rough W surface under 80° a huge discrepancy is demonstrated since the experimentally found backward reflection is not reproduced in the modeling. Both cases

which do not show a sufficient agreement between modeling and experiment show a common relation in the comparison of the average pathlengths. Comparing the pathlength of one angle of incidence and different surface roughness, the 30° smooth and 80° rough cases have a greater pathlength and more collisions in a collision cascade compared to the other topology.

## 5. Conclusions

The effect of the W surface roughness on the angular distribution of reflected and sputtered $^{13}C^+$ is investigated experimentally and numerically. By choosing $^{13}C^+$ projectiles, the natural abundance of 1.11% compared to $^{12}C$ is used in these experiments. The cleaning process of the Ti foil eliminates the impurities on the surface descending from the manufacturing process and from air exposure. The amount of $^{13}C$ is reduced by a factor of 10.

It is confirmed that sputtering leads to a cosine angular distribution, whereas reflection with projectile energies of 950 eV is demonstrated to be mainly specular. The surface roughness induces a change in the effective particle and energy reflection coefficients and sputtered fractions due to the fluence dependent change of the elemental composition of the surface (e.g., as seen in Figure 8b). In this study, it is shown that a higher surface roughness leads to an increase in absolute values of the effective reflection coefficient and sputtered fractions as the angle of incidence with respect to the surface normal is increased. Furthermore, an enhancement of $^{13}C$ sticking for large angles of incidence is demonstrated experimentally: A five times higher $^{13}C$ deposition on W was measured after the ion bombardment of the rough W surface under an angle of incidence of 80°. In the experiment we found differences of the deposited $^{13}C$ amount on W and the angular distribution of reflected and sputtered $^{13}C$ are likely to be of a geometrical origin. Particles impinging at a shallow angle to the surface almost directly get caught in the morphology surrounding the impingement point. Particles which are directly reflected correspond to a bombardment at 10° to the surface normal of a smooth surface which explains the experimentally found angular distribution in the 80° rough case. Performed simulations demonstrate the experimental trend of a higher deposition on the rough surface compared to the smooth surface by predicting a 1.19 higher deposition on the rough sample but show an angular distribution of projectiles which corresponds to forward scattering. The codes based on the BCA neither predict the experimental angular distributions in the given energy range and projectile-target combination nor reproduce the higher effective sticking probability for an angle of incidence of 80°. More complex simulations in 3D with a 3D surface topology are required here.

It is found that the angular distribution of reflected particles is correlated with the number of collisions in a collision cascade. For specular reflection, the collision cascade has a significantly lower number of collisions in a cascade. In addition, the momentum of the incoming particle parallel to the surface is better conserved when the collision cascade has a smaller number of collisions.

It is demonstrated that the surface roughness significantly alters the ion-surface interaction processes. The findings of this study not also play a role for fusion devices, but can also be used for vapor deposition methods. For future fusion devices such as ITER and DEMO, these findings are essential for codes which treat plasma-wall interaction processes. The codes need to consider the roughness induced effects. Otherwise consecutive reflection processes could lead to energy distributions of reflected particles which are shifted to too low energies compared to experimental energy distributions, since less collisions in a collision cascade result in higher particle energies of reflected particles. Thus, iterative erosion and deposition calculations could obtain a significant systematic error.

These implications lead to the need to further characterize surface roughness effects on ion-surface interaction processes with additional dedicated experiments. Future experiments should be accompanied by simulations with a complexity beyond the BCA and with a 3D surface morphology e.g., with molecular dynamics simulations. These studies should address the correlation of different roughness parameters with the effective sticking probability with respect to the angle of incidence of the projectile. It should be investigated if a critical threshold of the roughness is existent and the

significance of its influence on surface interaction processes. Furthermore, the influence of different roughness parameters on the reflection and its angular distribution should be studied, such as the skewness, which gives a measure of the asymmetry of the surface topology or the kurtosis which is a measure of the sharpness of the surface topology. Additionally, it is interesting to investigate if the surface morphology changes over the bombardment time and particle fluence, respectively. Since the present study shows a proof of principle of the effect for one projectile-target combination, a huge parameter space yet needs to be investigated. The impact of the mass ratio between the projectile and target atoms (reduced mass) in correlation with the angle of incidence as well as charge effects also should be carefully investigated to elucidate the underlying physics.

**Author Contributions:** Conceptualization: M.H., M.K., H.R.K. and A.L.; Methodology: M.H.; Software: M.H. and K.S.; Validation: M.H., M.K., H.R.K., K.S., C.S. and R.A.D.S.; Formal analysis: M.H.; Investigation: M.H., M.K., A.H., H.R.K. and C.S.; Resources: H.R.K. and R.A.D.S.; Data curation: M.H., M.K.; Writing—original draft preparation: M.H.; Writing—review and editing: M.H., M.K., H.R.K., A.L., K.S. and R.A.D.S.; Visualization: M.H.; Supervision: H.R.K. and A.L.; Project administration: M.H. and H.R.K.; Funding acquisition: A.L.

**Funding:** This research received no external funding.

**Acknowledgments:** The ion-surface interaction experiments (ALI) were performed at the Institute for Energy and Climate Research: Plasma Physics at the Forschungszentrum Jülich GmbH. The NRA measurements were performed at the Peter-Grünberg-Institut - Institute of Semiconductor Nanoelectronics at the Forschungszentrum Jülich GmbH. The SIMS measurements were performed at the Institute for Physical Chemistry, RWTH Aachen University. This study is an excerpt of the Phd thesis "Influence of 3D geometry and surface roughness on plasma-wall interaction processes on tungsten surfaces" by Maren Hellwig at the Ruhr University Bochum and at the Institute for Energy and Climate Research: Plasma Physics at the Forschungszentrum Jülich GmbH. She gratefully thanks the referees of her Phd thesis Christian A. E. Linsmeier and Achim von Keudell.

**Conflicts of Interest:** The authors declare no conflict of interest.

## Abbreviations

The following abbreviations are used in this manuscript:

| | |
|---|---|
| AFM | Atomic Force Microscopy |
| BCA | Binary collision approximation |
| NRA | Nuclear reaction analysis |
| SIMS | Secondary ion mass spectrometry |
| UHV | Ultra-high vacuum |

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
