# Peer review of "Impact of Surface Roughness on Ion-Surface Interactions Studied with Energetic Carbon Ions 13C+ on Tungsten Surfaces"

_condensedmatter, doi:10.3390/condmat4010029_

Round 1

Reviewer 1 Report

This manuscript reports a complete and very interesting research work on the role of the surface status in the interaction between energetic ions and substrate in view of particular application in nuclear fusion reactors. As a consequence particular projectile (C) and target (W) are considered. An original algorithm (based on the visibility concept) is presented to take into account the role of the surface morphology when collision are simulate by a BCA code, which instead assumes a flat interface. Experiment are designed and realized to derive angular distribution of emerging particles which can be used to test the simulations. The discrepancy between simulations and experimental results are strong. Anyhow, I believe that this is not a limit of the paper since it simply show the necessity of improvement of the modelling approach. Moreover, the data derived can be really useful for calibration of the numerical codes when the particular role of the surface has to be addressed. I would suggest the publication of this manuscript after three (minor) modifications. 1) In the presentation of the approach it is not fully clear if a real (planar) 3D roughness is modelled or if some rotational symmetry is assumed. In the latter case the eventual approximation in the evaluation of the visibility effects should be discussed. 2) The discontinuity of the energy distribution at 700eV in the figure 10 is noticed but it is not related to the calibration of the BCA computation. I believe that the authors should check the calibration (and related thresholds) of the energy loss mechanisms implemented in the BCA to understand this feature. 3) Due the strong discrepancy observed between BCA analysis and experimental data the use of accurate Molecular Dynamics simulations should be suggested in the conclusions (Chapter 5) for further investigation on this subject.

Author Response

Answer to review report 1 (Review date 06-02-2019)

Dear reviewer,

Thank you very much for reviewing our manuscript. 

1) Your remark: In the presentation of the approach it is not fully clear if a real (planar) 3D roughness is modelled or if some rotational symmetry is assumed. In the latter case the eventual approximation in the evaluation of the visibility effects should be discussed.

The roughness was implemented by taking the several 2D line scans of the AFM scan and compute the angular distributions of incoming particles on the experimental rough surface by a rotation. No difference was seen for the simulation with the different line scans since they were very similar. The 3D results of the SDTrimSP output were extracted which correspond to the 3D experimental interval. The following was added to the manuscript to clarify the simulation method: 

Line 85-91: „The surface roughness measurements were performed in three dimensions (3D). Several two dimensional (2D) line scans were used for the calculation of the normalized angular distribution of incidence angles. It was seen, that the measured surface was similar in such a way, that the normalized angular distributions of incidence angles of different line scans were very similar. Thus, the surface roughness can only be approximated with SDTrimSP by considering an angular distribution of incoming particles in contrast to one incidence angle for smooth surfaces. The present procedure takes a 3D problem, approximates it in 2D and calculates a 3D output.“ 

In lines 106-115 further clarifications to the use of 3D and 2D data is given and it is pointed out that 3D results of the simulations and 3D results of the experiments are compared. 

2) Your remark: The discontinuity of the energy distribution at 700 eV in the figure 10 is noticed but it is not related to the calibration of the BCA computation. I believe that the authors should check the calibration (and the related thresholds) of the energy loss mechanisms implemented in the BCA to understand this feature. 

This investigation of the artifact is very important and the authors agree with the reviewer that the implemented energy loss mechanisms are most likely responsible for the discontinuity. For this investigation simulations with the MARLOWE code were performed and as described show the same discontinuity. Unfortunately a further investigation is out of the scope of this study. Thus, the following was added to the article (lines 326-328):

„The reason for the artifact needs to be investigated further and is most likely generated by the implemented energy loss approximations in both codes. This investigation is beyond the scope of the presented article but must be pursuit further.“

3) Your remark: Due to the strong discrepancy observed between BCA analysis and experimental data the use of accurate Molecular Dynamics simulations should be suggested in the conclusions (Chapter 5) for further investigation of this subject. 

Thank you very much for this recommendation. The suggestion was added to Chapter 5 in lines 390 - 393: 

„These implications lead to the need to further characterize surface roughness effects on ion-surface interaction processes with additional dedicated experiments. Future experiments should be accompanied by simulations with a complexity beyond the binary collision approximation and with a 3D surface morphology e.g. with molecular dynamics simulations.

The authors would gratefully like to thank you very much for your thoughtful comments, they were highly appreciated. 

Reviewer 2 Report

It is an interesting study on the effect of surface roughness on angular distributions of reflected and physically sputtered particles. The case study is the bombardment of C+ ions with energy of 950 eV on W with incident angles of 30o and 80o; the incident angles refer to the mean plane of the surfaces. The study includes both measurements [secondary ion mass spectrometry (SIMS) and nuclear reaction analysis (NRA)] and simulation of ion surface interactions with the SDTrimSP code. A code for the pre- and post-processing of the SDTrimSP results has been developed. A comparison of measurements and simulation results are performed. The experimental as well as simulation results show that the impact of surface roughness is significant on the angular distribution of reflected and sputtered particles. It is demonstrated that the effective sticking of C on W is a function of the angle of incidence and surface morphology.

This will be suitable for publication in Condens. Matter after changes addressing the following points:

1) What is the critical value of roughness (e.g. rms value) above which the roughness affects the effective sticking probability? What are the factors affecting this critical value? A discussion on this critical value would be very useful for the reader.

2) The authors should clarify whether the calculations with SDTrimSP are performed in 2D, i.e. by using only one AFM scan, or in 3D, i.e. by using the 3D surface morphology coming from the AFM. In case they use a 2D scan, do they expect any effect on the conclusions? The results of SDTrimSP are in 2D while the results of the experiment correspond to 3D.  

3) The results of SDTrimSP demonstrate the trends of the measurements, however, there are discrepancies in the absolute values (e.g. on the deposited 13C on the rough and the smooth surface at incident angle of 80o). What is the origin of these discrepancies? How do the authors think to improve the model and diminish these discrepancies with the measurements?

4) There must an error in the values of rms roughness in Table 1. The values are in nm and not μm, aren’t they?

5) Minor changes:

a) Page 3, Line 85: Please substitute “are” with “is”.

Author Response

Answer to review report 2 (Review date 07-02-2019)

Dear reviewer,

Thank you very much for reviewing our manuscript. 

1) Your remark: What is the critical value of roughness (e.g. rms value) above which roughness affects the effective sticking probability? What are the factors affecting this critical value? A discussion on this critical value would be very useful for the reader. 

This very interesting question but unfortunately cannot be answered with the present experimental study. Thus, the following text was added to the Conclusions starting at line 390:

„These implications lead to the need to further characterize surface roughness effects on ion-surface interaction processes with additional dedicated experiments. Future experiments should be accompanied by simulations with a complexity beyond the binary collision approximation and with a 3D surface morphology e.g. with molecular dynamics simulations.

These studies should address the correlation of different roughness parameters with the effective sticking probability with respect to the angle of incidence of the projectile. It should be investigated if a critical threshold of the roughness is existent and the significance of its influence on surface-interaction processes. Furthermore, the influence of different roughness parameters on the reflection and its angular distribution should be studied like the skewness which gives a measure of the asymmetry of the surface topology or the kurtosis which is a measure of the sharpness of the surface topology. Additionally, it is interesting to investigate if the surface morphology changes over the bombardment time and particle fluence, respectively. Since the present study shows a proof of principle of the effect for one projectile-target combination, a huge parameter space yet needs to be investigated. The impact of the mass ratio between the projectile and target atoms (reduced mass) in correlation with the angle of incidence as well as charge effects also should be carefully investigated in order to elucidate the underlying physics.“

2) Your remark: The authors should clarify whether the calculations with SDTrimSP are performed in 2D, i.e.by using only one AFM scan, or in 3D, i.e by using the 3D surface morphology coming from the AFM . In case they use a 2D scan, do they expect any effect on the conclusions? The results of SDTrimSP are in 2D while the results of the experiment correspond to 3D.

The SDTrimSP simulations were performed in 3D. The roughness was implemented by taking the several 2D line scans of the AFM scan and check if it was representative by computing the angular distributions of incoming particles on the experimental rough surface by a rotation and are thus in 2D. No difference was seen for the normalized angular distribution of incoming particles which is used as input for SDTrimSP. The simulation results are in 3D. The simulation results which correspond to the 3D experimental interval were taken for comparison. The following was added to the manuscript to clarify the simulation method: 

Line 85-91: „The surface roughness measurements were performed in three dimensions (3D). Several two dimensional (2D) line scans were used for the calculation of the normalized angular distribution of incidence angles. It was seen, that the measured surface was similar in such a way, that the normalized angular distributions of incidence angles of different line scans were very similar. Thus, the surface roughness can only be approximated with SDTrimSP by considering an angular distribution of incoming particles in contrast to one incidence angle for smooth surfaces. The present procedure takes a 3D problem, approximates it in 2D and calculates a 3D output.“ 

In lines 106-115 further clarifications to the use of 3D and 2D data is given and it is pointed out that 3D results of the simulations and 3D results of the experiments are compared. 

Another addition to the text was done from line 337-338 and discusses: 

„A possible reason for the discrepancies is the approximation of a 3D surface morphology to a 2D problem. Thus, 3D simulations should be performed in future studies.“

3) Your remark: The results of the SDTrimSP demonstrate the trends of the measurements, however, there are discrepancies in the absolute values (e.g. on the deposited 13C on the rough and the smooth surface at incident angles of 80°). What is the origin of these discrepancies? How do the authors think to improve the model and diminish these discrepancies with the measurement?

The authors added to the conclusion following paragraph beginning in line 367:

„In the experiment we found differences of the deposited 13C amount on W and the angular distribution of reflected and sputtered 13C are likely to be of a geometrical origin. Particles impinging at a shallow angle to the surface almost directly get caught in the morphology surrounding the impingement point. Particles which are directly reflected correspond to a bombardment at 10° to the surface normal of a smooth surface which explains the experimentally found angular distribution in the 80° rough case. Performed simulations demonstrate the experimental trend of a higher deposition on the rough surface compared to the smooth surface by predicting a 1.19 higher deposition on the rough sample but show an angular distribution of projectiles which corresponds to forward scattering. The codes based on the binary collision approximation neither predict the experimental angular distributions in the given energy range and projectile target combination nor reproduce the higher effective sticking probability for an angle of incidence of 80°. More complex simulations in 3D with a 3D surface topology are required here.“

Furthermore in lines 391-393 it is suggested to perform molecular dynamics simulations:

Future experiments should be accompanied by simulations with a complexity beyond the binary collision approximation and with a 3D surface morphology e.g. with molecular dynamics simulations.“

4) The reviewer pointed out, that the unit of the rms roughness could be wrong: After careful recalculation we found, that the wrong data is displayed in this table. The error was corrected and every parameter is displayed in nanometers. Thank you very much for finding this mistake!

5) The reviewer asked to substitute „are“ with „is“ on page 3, line 85: The sentence „AFM scans of each sample surface are performed and respective angular distribution of incoming particles are computed“ was changed to „AFM scans of each sample surface are performed and respective angular distributions of incoming particles are computed.

The authors would gratefully like to thank you very much for your thoughtful comments, they were highly appreciated. 

Reviewer 3 Report

The topic is interesting, however the roughness and surface analysis is unclear:

As you study the surface and roughness, the AFM images of polished/rough surfaces before and AFTER the treatment must be shown at the appropriate scale. Perhaps, you will see some details to descuss.

At micro- submicro-scale the roughness depends on the size of the area. Therefore, the choise of 100x100 micron area (as can be imagined from Fig. 5) must be explained. If the ions interact with some part of the surface, for example they prefer depressions rather than slopes (this can be shown by AFM analysis of treated sumples) than the parameter roughness could be reconsidered.

Roughness is the simpliest parameter. Perhaps, skewness or curtosis would give more information.

Image at Fig. 4b is out of focus.

Author Response

Answer to review report 3

Dear reviewer,

Thank you very much for reviewing our manuscript.

1)Your remark: As you study the surface and roughness, the AFM images of polished/rough surfaces before and after the treatment must be shown at the appropriate scale. Perhaps, you will see some details to discuss.

This is a very interesting approach. The priority of this study was the analysis of the Ti sample by SIMS and the NRA analysis of the W sample after exposure. Unfortunately AFM scans of these samples after exposure were not performed due to the very extensive analysis procedures after SIMS and NRA measurements. In a further study it is of key importance to investigate the surface morphology changes with bombardment time and particle fluence, respectively. Line 398-399:

Additionally, it is interesting to investigate if the surface morphology changes over the bombardment time and particle fluence, respectively.“ 

2) Your remark: At micro- and submicroscopic-scale the roughness depends on the size of the area. Therefore, the choice of 100x100 micron area (as can be imagined from Fig. 5) must be explained. If the ions interact with some part of the surface, for example they prefer depressions rather than slopes (this can be shown by AFM analysis’s of treated samples) than the parameter roughness could be reconsidered.

The chosen size of the AFM sample scan area was given by the maximum possible scanning area of our AFM device. It was found that the given scans were representative for all samples. Thus, it was shown that the sample preparation was performed very thorough. The following was added to the paper in line 144 and following:

The surface topology is measured with an Atomic Force Microscope (AFM) on 100 ?m x 100 ?m areas, which was the maximum possible scanning area of the utilized AFM. The scanned areas represent the sample morphology. For the simulations, AFM measurement results of lines scans are used and are depicted in Figure 5.

Regarding your comment, that ions could interact preferably with special areas of the sample, the following was added from line 118 on: 

„Investigations of the influence of the surface morphology on the angular distribution of reflected and sputtered particles are performed with C+ ions impinging W surfaces at 950 eV. Projectiles at the given energy will uniformly bombard the sample surface in contrast to projectiles with energies close to the surface binding energy. It the latter case, the surface morphology with its slopes and sinks could affect the interaction process more strongly, which needs to be investigated in a different study.“

3) Your remark: Roughness is the simplest parameter. Perhaps, skewness or curtosis would give more information. 

The authors would like to thank the reviewer for this recommendation. It was implemented in the paper in the conclusion (396-398):

„Furthermore, the influence of different roughness parameters on the reflection and its angular distribution should be studied like the skewness which gives a measure of the asymmetry of the surface topology or the kurtosis which is a measure of the sharpness of the surface topology.“   

4) Your remark: Image at Fig. 4b is out of focus

The image in Figure 4b is partially out of focus. The depth of field of the microscope was not large enough to show the surface topology completely focus. Therefore, the focus was chosen such, that most of the sample surface is within the depth of field.

The authors would gratefully like to thank you very much for your thoughtful comments, they were highly appreciated. 

Round 2

Reviewer 3 Report

1. The authors doesn't understand that roughness, lets say R, strongly depends on the chosen scale:

if you measure 1x1 um area you will get R1, the area 10x10 um will give you R2 > R1....

So, what is the physical meaning of the selection of 100x100 um? For me, this is quite large area in the scope of the ions interactions with the surface.

Moreover, what is the resolution of your 100um AFM-profile? This is unclear.

For example, if you performed 256x256 scan, then the distance between the measured profile points is 0.39 um. A lot of data is missing.

Again, this is not the scale of ions interactions with the surface.

2. The AFM of SEM imaging of the surfaces before and after the measurements must be performed and shown.

Otherwise, all the talks about the surface and the roughness and the following simulations are only the speculations.

Author Response

Answer to review report 3 (Review date 19-02-2019)

Dear reviewer,

thank you for your comments.

1) Your remark: The authors doesn't understand that roughness, lets say R, strongly depends on the chosen scale:

if you measure 1x1 um area you will get R1, the area 10x10 um will give you R2 > R1....

So, what is the physical meaning of the selection of 100x100 um? For me, this is quite large area in the scope of the ions interactions with the surface.

We are well aware, that roughness parameters can be dependent on the size of the chosen area.

However, our main interest for this study with respect to the AFM measurements is the distribution of surface angles. For this reason, we used the largest scanning area of our AFM device of 100 µm x 100 µm to have the longest line scans available with the most data points to achieve the best possible statistics. It was ensured by other test measurements that the chosen line scan is representative for the whole sample. All tests gave very similar results with respect to the roughness profiles in the line scans. Comments on this topic were added in revision round one to lines 87-88.

The parameters R_RMS, R_10 and R_a are not used for further calculations but are used for the best possible description of the surface. In addition, we depicted the used roughness profiles (line scans) of the rough and smooth surface in Fig. 5 to clearly describe our surfaces.

Your remark: Moreover, what is the resolution of your 100um AFM-profile? This is unclear.

For example, if you performed 256x256 scan, then the distance between the measured profile points is 0.39 um. A lot of data is missing.

The resolution of our AFM measurements is 195 nm which equals to 512 x 512 measured points. We added this information to lines 145-146. We are not aware of any missing data. Please specify, which data you think is missing.

Your remark: Again, this is not the scale of ions interactions with the surface.

Since the fluence of the 13C ions which bombard the W surface is in the range of 1.7 - 10E21m-2 and the spot size of the ion beam on the W sample is between 0.43 mm^2 (for 30 degrees) and 3.99 mm^2 (for 80 degrees), the performed experiments aim at the statistical analysis of the ion-surface interactions. Consequently, a statistical analysis of the ion-surface interactions with described interaction processes was performed in 3D with the described simulation procedure. The actual angles between incoming particle and surface are calculated with the measured surface morphology. These angles are used as start parameters for the simulations.

No roughness parameter was needed for the simulations. Furthermore, Fig. 5 shows, that the scale and the resolution of the AFM scan is sufficient for the given experiments.

2) Your remark: The AFM of SEM imaging of the surfaces before and after the measurements must be performed and shown.

Otherwise, all the talks about the surface and the roughness and the following simulations are only the speculations. 

The images of the sample surface are recorded with an optical microscopy, not with a SEM. This was clarified in the caption of Fig. 4.

The focus of this study is the investigation of incoming, deposited, reflected and sputtered particles on smooth as opposed to technical rough surfaces. This particular study does not aim to investigate the evolution of surface roughness during ion bombardment. However, this is would be an interesting topic for further studies.

Moreover, the presented experiments have a 13C fluence of 1E21m-2 at maximum which corresponds to 1E17cm-2 (Fig.8). With a W sputter yields of 0.4, 4E16cm-2 W atoms are sputtered. This corresponds to roughly 40 monolayers and 12 nm. Thus, the surface roughness profile of the rough surface as shown in Fig. 5 will not significantly change. For the smooth surface no significant effect on its roughness profile is to be expected, since irradiation is uniform. An AFM scan of the surfaces after exposure is not essential for the presented study.

Round 3

Reviewer 3 Report

You done much work.

However its all is unreliable without the explanation of the choise of roughness scale and the microstructure of the surfaces.